# Prognostic Value of Transaminases and Bilirubin Levels at Admission to Hospital on Disease Progression and Mortality in Patients with COVID-19—An Observational Retrospective Study

**DOI:** 10.3390/pathogens11060652

**Published:** 2022-06-06

**Authors:** Antonio Russo, Mariantonietta Pisaturo, Roberta Palladino, Paolo Maggi, Fabio Giuliano Numis, Ivan Gentile, Vincenzo Sangiovanni, Vincenzo Esposito, Rodolfo Punzi, Giosuele Calabria, Carolina Rescigno, Angelo Salomone Megna, Alfonso Masullo, Elio Manzillo, Grazia Russo, Roberto Parrella, Giuseppina Dell’Aquila, Michele Gambardella, Antonio Ponticiello, Nicola Coppola

**Affiliations:** 1Infectious Diseases Unit, Department of Mental Health and Public Medicine, University of Campania “L. Vanvitelli”, 80138 Naples, Italy; antoniorusso.ar.ar@gmail.com (A.R.); mariantonietta.pisaturo@unicampania.it (M.P.); roberta.palaldino@studenti.unicampania.it (R.P.); 2Infectious Diseases Unit, A.O. S Anna e S Sebastiano, 81100 Caserta, Italy; paolo.maggi@unicampania.it; 3Emergency Unit, PO Santa Maria delle Grazie, 80078 Pozzuoli, Italy; fabiogiuliano.numis@aslnapoli2nord.it; 4Infectious Disease Unit, University Federico II, 80138 Naples, Italy; ivan.gentile@unina.it; 5Third Infectious Diseases Unit, AORN dei Colli, PO Cotugno, 80131 Naples, Italy; sangio.vincenzo@gmail.com; 6IV Infectious Disease Unit, AORN dei Coli, PO Cotugno, 80131 Naples, Italy; vincenzoesposito@ospedalideicolli.it; 7Hepatic Infectious Disease Unit, AORN dei Colli, PO Cotugno, 80131 Naples, Italy; rodolfo.punzi@ospedalideicolli.it; 8IX Infectious Disease Unit, AORN dei Coli, PO Cotugno, 80131 Naples, Italy; g.calabria@tin.it; 9First Infectious Disease Unit, AORN dei Coli, PO Cotugno, 80131 Naples, Italy; carolinarescigno@libero.it; 10Infectious Diseease Unit, A.O. San Pio, PO Rummo, 82100 Benevento, Italy; angelo.salomonemegna@ao-rummo.it; 11Infectious Disease Unit, A.O. San Giovanni di Dio e Ruggi D’Aragona, 84131 Salerno, Italy; al.masullo@alice.it; 12VIII Infectious Disease Unit, AORN dei Coli, PO Cotugno, 80131 Naples, Italy; manzillo@libero.it; 13Infectious Disease Unit, Ospedale Maria S.S. Addolorata di Eboli, ASL, 84025 Salerno, Italy; gr.russo@aslsalerno.it; 14Respiratory Infectious Diseases Unit, AORN dei Colli, PO Cotugno, 80131 Naples, Italy; roberto.parrella@ospedalideicolli.it; 15Infectious Diseases Unit, AO, 83100 Avellino, Italy; dellaquilagiuseppina@libero.it; 16Infectious Diseease Unit, PO S. Luca, Vallo della Lucania, ASL, 84078 Salerno, Italy; gambardella1960@gmail.com; 17Pneumology Unit, AORN, 81100 Caserta, Italy; antonio.ponticiello@unina.it

**Keywords:** transaminases, bilirubin, COVID-19, SARS-CoV-2 infection, severity of disease, mortality

## Abstract

Introduction: Given the impact of COVID-19 on the world healthcare system, and the efforts of the healthcare community to find prognostic factors for hospitalization, disease progression, and mortality, the aim of the present study was to investigate the prognostic impact of transaminases and bilirubin levels at admission to hospital on disease progression and mortality in COVID-19 patients. Methods: Using the CoviCamp database, we performed a multicenter, observational, retrospective study involving 17 COVID-19 Units in southern Italy. We included all adult patients hospitalized for SARS-CoV-2 infection with at least one determination at hospital admission of aminotransaminases and/or total bilirubin. Results: Of the 2054 patients included in the CoviCamp database, 1641 were included in our study; 789 patients (48%) were considered to have mild COVID-19, 347 (21%) moderate COVID-19, 354 (22%) severe COVID-19, and 151 patients (9%) died during hospitalization. Older age (odds ratio (OR): 1.02; 95% confidence interval (CI) 1.01–1.03), higher Charlson comorbidity index (CCI) (OR 1.088; 95%CI 1.005–1.18), presence of dementia (OR: 2.20; 95% CI: 1.30–3.73), higher serum AST (OR: 1.002; 95% CI: 1.0001–1.004), and total bilirubin (OR: 1.09; 95% CI: 1.002–1.19) values were associated with a more severe clinical outcome. Instead, the 151 patients who died during hospitalization showed a higher serum bilirubin value at admission (OR 1.1165; 95% CI: 1.017–1.335); the same did not apply for AST. Discussion: Patients with COVID-19 with higher levels of AST and bilirubin had an increased risk of disease progression.

## 1. Introduction

Since December 2019 to date, the novel Coronavirus Disease 2019 (COVID-19) has spread all over the world, and more than 430,000,000 cases and more than 5,000,000 deaths have been registered [1].

A wide range of clinical manifestations and outcomes of COVID-19 have been reported, from asymptomatic and mild respiratory infections to pneumonia and acute respiratory distress syndrome (ARDS), as well as life-threatening multiple organ failure [2]. Several studies have highlighted the impact of different risk factors, such as being aged over 65, diabetes, hypertension, chronic kidney disease, dementia, and active cancer, all of which contribute to increased mortality and need for non-invasive and invasive ventilation [3,4,5,6,7]. The identification of clinical and biochemical factors potentially associated with severe COVID-19 is of paramount importance, and may help healthcare professionals to personalize treatment, especially in the current era of early effective treatment, i.e., antivirals and monoclonal antibodies, according to the risk factors of each subject, as well as to allocate proper resources at all levels of care.

Although the lung is the main target of COVID-19, other organs, such as the liver, represent direct or indirect targets of the virus [8]. Liver damage is likely multifactorial: it may be due not only to direct viral damage, but also due to a systemic cytokine storm, pro-thrombotic state, drug use and abuse during COVID-19, and hypoxic injury [8]. Finally, several studies have investigated the role of liver enzymes on disease progression and mortality [8,9,10,11,12,13,14,15,16,17,18,19,20,21,22], showing a relation between the elevation of liver enzymes, considered as a manifestation of liver damage, and the severity of COVID-19 disease. In addition, the impact that the SARS-CoV-2 pandemic has determined on patients with liver disease has been considerable for both outpatient and inpatient management [23,24,25].

The aim of this observational, retrospective study on a large cohort of hospitalized COVID-19 patients was to assess the impact of aspartate aminotransferase (AST), alanine-aminotransferase (ALT), and total bilirubin levels at admission to hospital on disease progression and mortality.

## 2. Materials and Methods

### 2.1. Study Design and Setting

We performed a multicenter, observational, retrospective study involving 17 COVID-19 units in 8 cities in the Campania region of southern Italy: Naples, Caserta, Salerno, Benevento, Avellino, Pozzuoli, Eboli, and Vallo della Lucania. All adult (≥18 years) patients, hospitalized with a diagnosis of SARS-CoV-2 infection confirmed by a positive reverse transcriptase-polymerase chain reaction (RT-PCR) on a naso-oropharyngeal swab, from 28 February 2020, to 1 November 2021 at one of the centers participating in the study, were enrolled in the CoviCamp cohort. Exclusion criteria included minority age, and a lack of clinical data and/or of informed consent. All of the patients enrolled in the present study were symptomatic for COVID-19. No study protocol or guidelines regarding the criteria of hospitalization were shared among the centers involved in the study, and the patients were hospitalized following the decision of physicians of each center.

From the CoviCamp cohort, we included for the present study all patients for whom a determination at admission of AST and/or ALT and/or total bilirubin was available.

The study was approved by the Ethics Committee of the University of Campania L. Vanvitelli, Naples (n°10877/2020). All procedures performed in this study were in accordance with the ethics standards of the institutional and/or national research committee and with the 1964 Helsinki declaration and its later amendments or comparable ethics standards. Informed consent was obtained from all participants included in the study.

This study was reported following the STROBE recommendations for an observational study (Appendix A).

### 2.2. Data Collection

All demographic and clinical data and therapy details of patients with SARS-CoV-2 infection enrolled in the cohort were collected in an electronic database. From this database, we extrapolated the data for the present study.

### 2.3. Variables and Definitions

The microbiological diagnosis of SARS-CoV-2 infection was defined as a positive RT-PCR test via a naso-oropharyngeal swab, according to indications by the manufacturer (RT-PCR kit, Bosphore V3; Anatolia Genework, Turkey).

We divided the patients enrolled according to the clinical outcome of COVID-19 during hospitalization: mild outcome, moderate outcome, severe outcome, and death. Precisely, the patients with a mild infection did not need oxygen (O_2_) therapy and/or had a MEWS score below 3 points during hospitalization. The patients with a moderate infection were hospitalized, and required non-invasive O_2_ therapy (excluding high flow nasal cannula) and/or had a MEWS score equal to or above 3 points (≥3) during hospitalization. The patients with a severe infection needed management in an intensive care unit (ICU) and/or high flow nasal cannula or invasive/non-invasive mechanical ventilation during hospitalization. The patients were followed until SARS-CoV-2-RNA results were negative, tested via a naso-oropharyngeal swab, and/or discharged from hospital or died. The normal range of AST, ALT, and bilirubin was the same in all centers (AST 5–33 U/L, ALT 5–32 U/L, total bilirubin < 1.2 mg/dL).

### 2.4. Statistical Analysis

For the descriptive analysis, categorical variables were presented as absolute numbers and their relative frequencies. Continuous variables were summarized as mean and standard deviation if normally distributed, or as median and interquartile range (Q1–Q3) if not normally distributed. We performed a comparison of patients with mild disease, moderate disease, severe disease or who died using a chi square test for categorical variables, or ANOVA for continuous variables. We performed a comparison of patients who were discharged from hospital and those who died during hospitalization using a Pearson chi-square test or Fisher’s exact test for categorical variables, and Student’s *t*-test or a Mann–Whitney- or Kruskal–Wallis test for continuous variables. Odds ratios were calculated using binomial logistic regression or ordered logistic regression, when required; these analyses were performed only for parameters resulting as statistically significant through a univariate analysis. A *p*-value below 0.05 was considered statistically significant. Analyses were performed by STATA.

## 3. Results

During the study period, a total of 2054 patients with SARS-CoV-2 infection were included in the CoviCamp cohort. Considering the inclusion criteria for the present study, 1641 patients were included (Figure 1): the demographic and clinical data of the patients enrolled in the present study and those of the patients not enrolled were similar (Appendix A).

The demographic and clinical characteristics of the patients at enrolment are shown in Table 1. The majority (61%) of patients enrolled were males; the mean age was 62.3 ± 16.1 years, the Charlson comorbidity index (CCI) was 3.0 points (standard deviation SD 2.43), and the main co-morbidity reported was arterial hypertension (47.5%) (Table 1). Only 63 (3.8%) patients had a chronic liver disease (20 from a past hepatitis C virus infection, 9 from an active hepatitis C virus, 10 from a hepatitis B virus infection, 11 from alcohol use, 12 from non-alcoholic fatty liver disease, and 1 from cryptogenic aetiology).

During the period of study, the home therapy prescribed was substantially unchanged, excluding the first wave, which was poorly represented. The most important change during the study period was the compression of the uselessness of azithromycin therapy in the early stages, with the progressive discontinuation of its home use in the absence of signs of bacterial infection.

Regarding the laboratory analysis at admission, 27.7% of patients showed an abnormal serum value of AST, 23% of abnormal ALT values, and 12.6% of abnormal total bilirubin values.

Considering the severity outcome of COVID-19, 789 patients (48%) were considered to have mild, 347 (21%) moderate, 354 (22%) severe COVID-19 infection, and 151 patients (9%) died during hospitalization; all death was directly or indirectly related to COVID-19 (Table 2). Considering the demographic data which compared the four investigated groups, age was statistically different between the groups (mean age and SD: 59.31 ± 15.96, 61.34 ± 15.75, 63.18 ± 13.90, and 79.03 ± 12.03 years, respectively; *p* ≤ 0.001) (Table 2). With regard to the presence of co-morbidities, CCI and the presence of hypertension, cardiovascular disease, diabetes, chronic kidney disease, obstructive pulmonary disease, malignancy, and dementia showed a statistical difference among the groups (Table 2). Moreover, no statistical difference was present in the patients with chronic liver disease (*p* = 0.053) (Table 2). Considering the number of days under hospitalization (including death or discharge), there was a statistical difference between the groups (*p* ≤ 0.0001) (Table 2). Regarding liver laboratory data at admission, there was a statistical difference between the groups when analyzing AST, ALT, and bilirubin, shown as *p* < 0.0001, *p* < 0.0001, and *p* = 0.0172, respectively (Table 2), Moreover, analyzing AST levels according to intervals of altered values (group 1 = AST less than normal value (n.v.); group 2 = AST below “upper limit of normal” (ULN), ×1 ULN to ×2 ULN; group 3= AST from ×2 ULN to ×3 ULN; group 4 = from ×3 ULN to ×4 ULN; group 5 = AST from ×5 ULN to ×7 ULN, group 6 = AST more than ×7 ULN), the prevalence of patients with AST levels higher than 2× the normal value was higher in those with a severe outcome (Figure 2). Considering the high correlation between AST and ALT (0.886), and considering the trend, we excluded the ALT from logistic regression The ordered logistic regression confirmed that older age (odds ratio (OR): 1.02; 95% confidence interval (CI) 1.01–1.03), higher CCI (OR 1.088; 95%CI 1.005–1.18), presence of dementia (OR: 2.20; 95%: 1.30–3.73), and higher serum AST (OR: 1.002; 95% CI: 1.0001–1.004) and total bilirubin (OR: 1.09; 95% CI: 1.002–1.19) values were associated with a more severe clinical outcome (Table 2).

Including only the patients who had both AST and bilirubin determination at admission (1403 patients), we performed a comparison between the 1288 patients who were discharged alive from the hospital, and the 115 who died during hospitalization (Table 3). Compared with the patients who were discharged alive, those who died during hospitalization showed a higher mean age (*p* < 0.001), a higher Charlson comorbidity index (*p* < 0.001) (OR:1.303; 95% CI 1.192–1.424), and more frequently had cardiovascular disease (*p* = 0.015), diabetes (*p* < 0.001), chronic kidney disease (*p* < 0.001), chronic obstructive pulmonary disease (*p* < 0.001), malignancy (*p* = 0.009), or dementia (*p* < 0.001; Table 3). Considering the laboratory data, the mean AST and bilirubin serum levels were higher in the 115 patients who died (*p* = 0.003 and *p* < 0.0001, respectively; Table 3). The binary logistic regression confirmed that patients who died during hospitalization were older (OR: 1.115; 95% CI: 1.088–1.1142), had a higher CCI (OR: 1.303; 95% CI: 1.192–1.424), more frequently had dementia (OR: 4.332; 95% CI: 2.198–8.538), and a higher serum bilirubin value at admission (OR 1.1165; 95% CI: 1.017–1.335); the same did not apply for AST values (Table 3).

## 4. Discussion

The impact of COVID-19 on national healthcare systems worldwide has been outstanding [26,27]. Since the first wave, the need for prognostic factors that highlight the possibility of predicting a severe disease outcome have been the focus of several studies. This possibility may stratify the risk of a severe disease, reduce unnecessary hospitalizations, and allocate proper resources at all levels of care [3,28]. The most important impact on the reduction in hospitalizations in SARS-CoV-2 infection has been given by the large-scale use of vaccines since the beginning of 2021 [29], followed by the introduction of monoclonal antibodies and antivirals. However, considering the low prevalence of anti-SARS-CoV-2 vaccination in some countries, as well as the cost of monoclonal antibodies and antivirals, especially in developing countries, the scientific interest in identifying prognostic factors of severity of COVID-19 disease is still necessary and important.

In the present observational retrospective study performed in 19 COVID-19 units on a large sample of hospitalized patients in southern Italy, a direct independent correlation between higher serum values of bilirubin or AST at admission and disease outcome was demonstrated. In fact, considering the clinical outcome during hospitalization, the four disease groups identified showed a linear increase of bilirubin and AST levels. In addition, a higher serum value of bilirubin at admission to hospital independently correlated with the risk of death during hospitalization.

Our data are in line with other similar studies in the literature [12,13,14,15,16,17,18,19,20,21,22]. Those studies, mainly performed in the USA, showed a significant association between the increase in liver function and disease progression or death in patients with COVID-19 [20,21,22]. For example, Currier et al. investigated the impact of liver function tests (LFTs) on COVID-19 outcome in 8028 patients. Compared to COVID-19 patients without elevated LFTs, those with LFTs had a significantly higher probability of hospital admission, ICU admission, oro-tracheal intubation, and death (all *p* < 0.001) [21]. Rastogi et al. showed a significant upward trend in transaminases, alkaline phosphatase, prothrombin time, bilirubin, lactate dehydrogenase, and lipase and a downward trend in albumin with an increase in disease severity during COVID-19 hospitalization [20]. In addition, COVID-19-positive patients with hepato–pancreatic injury had a significantly higher mortality (OR 3.39, 95%CI 3.15–3.65) and longer hospital stay [20]. However, the samples of these studies included in most cases patients with a severe disease outcome (50–70%). In contrast, in the present study, which involves mainly infectious disease units, the prevalence of patients with a negative outcome did not exceed 30%, which appears to be closer to clinical practice. Moreover, in the present study, the evaluation of the association between LFTs and the severity of COVID-19 was performed on the values at baseline, with a greater significance in the stratification of risk progression.

The most common factors that may contribute to liver involvement in COVID-19 are a direct cytopathic effect, immune-mediated effects, hypoxia-induced changes, coagulopathy, right heart failure, and drug-induced liver injury. Considering direct cytopathic effects, it is widely known that angiotensin-converting enzyme 2 (ACE2) receptor provides a gateway for entry, and ACE2 is highly expressed on cholangiocytes; in contrast, it is low on hepatocytes, Kupfer cells, and endothelial cells. However, ACE2 expression is upregulated by many factors, including pre-existing liver disease, hypoxia, drug-induced liver injury, and inflammation. Considering the immune-mediated effect, an exaggerated inflammatory response, i.e., a cytokine storm, leads to lymphocyte activation, neutrophilia, and a paradoxical increase in C-reactive protein and inflammatory cytokines. In addition to increasing cytokines and the activity of the immune system, a cytokine storm causes systemic hypotension, microthrombi, abnormal coagulation, and endothelial dysfunction. In particular, microvascular thrombosis and endothelial dysfunction may cause hepatic damage due to ischemic injury, both by direct hepatic involvement and by intestinal or myocardial damage. The association between higher AST and total bilirubin at the baseline, but not ALT serum values, and the progression of COVID-19 identified in the present study may be an expression of hypo-perfusion of liver tissue. Thus, the subjects with elevated initial AST and total bilirubin levels, having a sub-clinical tissue hypo-perfusion, may be at high risk of a severe disease outcome. Moreover, the patients with a pre-existing chronic liver disease showed a higher risk of mortality rate compared to those without (32% vs. 8%) [8]. In our study, the prevalence of chronic liver disease was low (only 63 patients, 3.7%.); thus, it was not possible to evaluate the impact of chronic liver disease on the outcome of COVID-19 in the present cohort.

Our study shows some limits: first, the retrospective nature of the study; second, we evaluated only hospitalized patients and hospital mortality; third, some data that could improve the results are lacking; fourth, the absence of analysis of the impact of viral variants. However, we underline that the demographic and clinical characteristics of the patients were similar to those not included in the present study. The strengths of our study are the multicenter nature of the design and the size of the population.

In conclusion, bilirubin and AST levels at admission are directly correlated with in-hospital progression of COVID-19. In addition, the increase in bilirubin levels at time of hospital admission is significantly related to hospital mortality; however, also considering the minor association, further research is required to validate the data shown. Considering the need for tools and prognostic examinations that can help to predict the clinical progression of disease in COVID-19 patients, our study, with others recently published [8,9,10,11,12,13,14,15,16,17,18,19], could be a useful guide, using liver damage and liver function tests to highlight patients at risk of COVID-19 progression or to predict the severity of disease progression.

## Figures and Tables

**Figure 1 pathogens-11-00652-f001:**
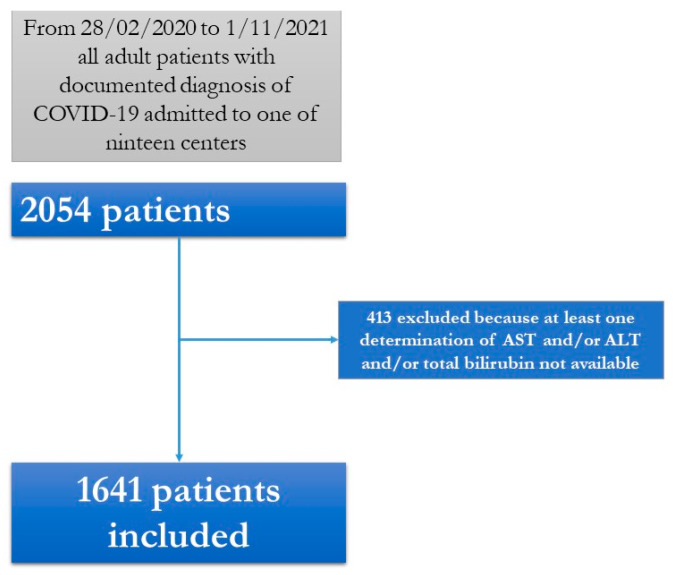
Flow chart of patients included.

**Figure 2 pathogens-11-00652-f002:**
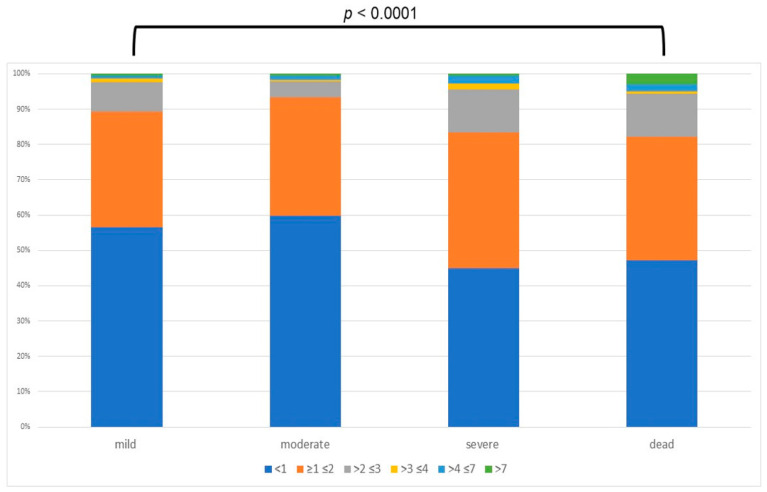
Prevalence of different cut-off of AST (below “upper limit of normal”) considering different clinical outcome.

**Table 1 pathogens-11-00652-t001:** Demographic, clinical and laboratory data of patients included.

**Males, N° (%)**	1003 (61%)
**Age, years, mean (SD)**	62.35 (16.14)
**Charlson comorbidity index, mean (SD)**	3 (2.43)
**N° (%) of patients with hypertension**	779 (47.5%)
**N° (%) of patients with cardiovascular disease**	467 (28.5%)
**N° (%) of patients with diabetes**	347 (21.1%)
**N° (%) of patients with chronic kidney disease**	147 (9%)
**N° (%) of patients with chronic obstructive pulmonary disease**	160 (9.8%)
**N° (%) of patients with chronic liver disease**	63 (3.8%)
**N° (%) of patients with malignancy**	112 (6.8%)
**N°(%) of patients with dementia**	90 (4.9%)
**Days from admission to discharge *, median (SD)**	15.64 (9.49)
**Mean (SD) white blood cells (WBC) at admission (cells/uL)**	9218.8 (8177.18)
**Mean (SD) international normalized ratio (INR) at admission**	1.14 (0.4)
**Mean (SD) Blood creatinine at admission (mg/dl)**	1.3 (1.8)
**Mean (SD) creatine phosphokinase (CPK) at admission (U/L)**	186 (460)
**Mean (SD) lacticodehydrogenase (LDH) at admission (U/I)**	359 (227)
**Mean (SD) PaO2/FiO2 Ratio (P/F) at admission**	239 (108)
**Mean(SD) ALT at T0 ** (U/L)**	52.38 (101.74)
**N° (%) of patients with abnormal serum ALT value ^$^**	386 (23%)
**Mean (SD) AST at T0 *** (U/L)**	46.05 (93.6)
**N° (%) of patients with abnormal serum AST value ^$$^**	455 (27.7%)
**Mean (SD) total bilirubin at T0 ^£^ (mg/dL)**	0.81 (1.76)
**N° (%) of patients with abnormal serum total Bilirubin value ^$$$^**	208 (12.6%)

* or death in patients who died during hospitalization; ** analysis performed on 1548 patients; *** analysis performed on 1445 patients; ^£^ analysis performed on 1403 patients; ^$^ Abnormal value defined as upper of 51 UI; ^$$^ Abnormal value defined as upper of 43 UI; ^$$$^ Abnormal value defined as upper of 1.00 mg/dL.

**Table 2 pathogens-11-00652-t002:** Demographic, clinical and laboratory differences at admission of patients grouped by disease severity.

							Multivariate Analysis Ordered Logistic Regression	
	Patients with Mild Clinical Outcome n=789 (48%)	Patients with Moderate Clinical Outcome n = 347 (21%)	Patients with Severe Clinical Outcome n = 354 (22%)	Patients Who Died during Hospitalization n = 151 (9%)	*p*-Value	*p*-Value for the trend	OR (95% CI)	*p*-Value
**Males, N. (%)**	485 (61)	204 (58,8)	232 (65,5)	80(53)	0.047 ^a^	0.559 ^d^	0.91 (0.72–1.14)	0.427
**Age, years, mean (SD)**	59.31 (15.961)	61.34 (15.752)	63.18 (13.896)	79.03 (12.030)	<0.001 ^c^	<0.0001 ^e^	1.02 (1.01–1.03)	<0.001
**Charlson comorbidity index, Median (Q1–Q3)**	2 (1–4)	2 (1–4)	3 (1–4)	5 (4–7)	<0.0001 ^b^	< 0.0001 ^e^	1.088 (1.005–1.18)	0.037
**N. (%) with hypertension**	341 (43.2)	173 (49.8)	180 (50.8)	86 (56.9)	0.004 ^a^	<0.0001 ^d^	1.07 (0.85–1.34)	0.563
**N. (%) with cardio-vascular disease**	201 (25.5)	87 (25)	100 (28.2)	79 (52.3)	<0.0001 ^a^	<0.001 ^d^	0.84 (0.63–1.10)	0.203
**N. (%) with diabetes**	152 (19.3)	68 (19.6)	73 (20.6)	54 (35.7)	<0.0001 ^a^	0.001 ^d^	0.99 (0.74–1.32)	0.931
**N. (%)with chronic kidney disease**	64 (8.1)	27 (7.7)	20 (5.6)	36 (23.8)	<0.0001 ^a^	0.001 ^d^	1.01 (0.67–1.53)	0.946
**N. (%)with chronic obstructive pulmonary disease**	62 (7.8)	31 (8.9)	37 (10.4)	30 (19.8)	<0.0001 ^a^	<0.001 ^d^	1.43 (0.99–2.08)	0.059
**N. (%)with chronic liver disease**	35 (4.4)	11 (3.1)	7 (1.9)	10 (6.6)	0.053 ^a^	0.718 ^d^	-	-
**N. (%) with malignancy**	55 (6.9)	18 (5.1)	21 (5.9)	18 (12)	0.044 ^a^	0.292 ^d^	0.89 (0.56–1.43)	0.632
**N.(%) with dementia**	29 (3,8)	7 (2)	12 (3,4)	32 (21,2)	<0.0001 ^a^	<0.001 ^d^	2.20 (1.30–3.73)	0.003
**Days from admission to discharge *, Median (Q1–Q3)**	13 (9–19)	15 (10–21)	18 (12–25)	10 (5–15)	<0.001 ^b^	0.437 ^e^	1.01 (0.99–1.02)	0.179
**Median (Q1–Q3) ALT at admission ****	32 (21–57)	29 (20–48)	37 (22–64)	24 (14.5–39)	<0.001 ^b^	0.314 ^e^	-	-
**Median (Q1–Q3) AST at admission *****	29 (20–46)	30 (21–41)	36 (25–50)	35 (24–54.5)	<0.0001 ^b^	<0.001 ^e^	1.002 (1.0001–1.004)	0.040
**Median (Q1–Q3) bilirubin at admission ^£^**	0.57 (0.4–0.8)	0.6 (0.44–0.83)	0.62 (0.46–0.89)	0.5 (0.4–0.9)	0.0172 ^b^	0.004 ^e^	1.09 (1.002–1.19)	0.045

* or death in patients who died during hospitalization; ** analysis performed on 1548 patients; *** analysis performed on 1445 patients; ^£^ analysis performed on 1403 patients; ^a^: Chi-Square test; ^b^: Kruskal–Wallis Test; ^c^: one-way ANOVA; ^d^: linear by linear association; ^e^: linear regression.

**Table 3 pathogens-11-00652-t003:** Demographic, clinical, and laboratory parameters in patients discharged from hospital and those who died during hospitalization (analysis performed on 1403 patients).

	Patients Discharged from Hospital n = 1288	Patients Who Died during Hospitalization n = 115	*p*-Value	Multivariate Analysis Ordered Logistic Regression	
				OR (CI 95%)	*p*-Value
**Males, N. (%)**	921 (71,5)	80(69.6)	0.034 ^a^	Not significant	0.666
**Age, years, mean (SD)**	60.70 (15.519)	79.03 (12.030)	<0.001 ^b^	1.115 (1.088–1.1142)	<0.001
**Charlson comorbidity index, Median (Q1–Q3)**	2 (1–4)	5 (4–7)	<0.001 ^b^	1.303 (1.192–1.424)	<0.001
**N. (%) with hypertension**	694 (53)	86 (74.8)	0.015 ^a^	Not significant	0.650
**N. (%) with cardio-vascular disease**	388 (30.1)	79 (68.7)	<0.0001 ^a^	Not significant	0.110
**N. (%) with diabetes**	293 (22.7)	54 (47)	<0.0001 ^a^	1.705 (1.037–2.801)	0.035
**N. (%) with chronic kidney disease**	111 (8.6)	36 (31.3)	<0.0001 ^a^	Not significant	0.61
**N. (%) with chronic obstructive pulmonary disease**	130 (10)	30 (26)	<0.0001 ^a^	Not significant	0.479
**N. (%) with chronic liver disease**	53 (4.1)	10 (8.6)	0.062 ^a^	-	-
**N. (%) with malignancy**	94 (7.3)	18 (15.6)	0.009 ^a^	Not significant	0.105
**N.(%) with dementia**	48 (3.7)	32 (27.8)	<0.0001 ^a^	4.332 (2.198–8.538)	<0.0001
**Days from admission to discharge *, Median (Q1–Q3)**	15(10–21)	10 (5–15)	<0.001 ^b^	0.933 (0.908–0.959)	<0.001
**Median (Q1–Q3) ALT at admission**	32 (21–55.5)	24 (14.5–39)	0.055 ^b^	-	-
**Median (Q1–Q3) AST at admission**	31 (21–46)	35 (24–54.5)	0.003 ^b^	Not significant	0.063
**Median (Q1–Q3) bilirubin at admission**	0.6 (0.42–0.82)	0.5 (0.4–0.9)	<0.0001 ^b^	1.1165 (1.017–1.335)	0.027

* or death in patients who died during hospitalization, ^a^: Chi-Square test; ^b^: two-sample *t*-test.

## Data Availability

Not applicable.

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
