# Peer review of "Prognostic Value of Transaminases and Bilirubin Levels at Admission to Hospital on Disease Progression and Mortality in Patients with COVID-19—An Observational Retrospective Study"

_pathogens, 2022, doi:10.3390/pathogens11060652_

Round 1

Reviewer 1 Report

The paper presents multicentre observational retrospective study about the role of laboratory values of liver enzymes to predict the disease course and mortality in hospitalised patients with COVID19 disease.

Could authors comment on the role of possible baseline liver disease or concomitant ambulatory treatment on the prevalence of increased baseline liver enzymes.

What is the median/mean time since the onset of COVID19 disease in which laboratory tests were performed? Is there any difference in time from the onset of disease and laboratory analysis between the examined groups?

Since the study did not include all COVID19 patients, specifically those not scheduled for hospitalization, could authors comment on the implications of the results of their study on the everyday clinical practice?

The title, abstract, manuscript organization, discussion, tables and references are appropriate.

Language requires some polishing to achieve precision, clarity and grammatical correctness.

Author Response

Dear Editor,

We re-submit our paper “Prognostic value of transaminases and bilirubin levels at admission to hospital on disease progression and mortality in patients with covid-19 - an observational retrospective study (tracking number: pathogens-1704753), modified according to the suggestions of the Editor and of the Reviewers.

POINT-BY-POINT ANSWER TO THE COMMENTS OF THE REVIEWER 1

Point 1: Could authors comment on the role of possible baseline liver disease or concomitant ambulatory treatment on the prevalence of increased baseline liver enzymes.

Answer: Thanks for the suggestions. In the discussion section of the new manuscript, we have discussed this point

Point 2: What is the median/mean time since the onset of COVID19 disease in which laboratory tests were performed? Is there any difference in time from the onset of disease and laboratory analysis between the examined groups?

Answer: of the 1,641 patients included in the present study, only for 766 patients the date of onset of symptoms attributable to COVID-19 was available, without difference in the patients with different outcome: 6.43 (+4.67) days in the 343 with mild outcome, 6.78 (+4.1) in the 176 with moderate outcome, 7.35 (+4,18) in the 160 with severe outcome, and 5.90 (+4.26) in the 87 who dead.

Point 3: Since the study did not include all COVID19 patients, specifically those not scheduled for hospitalization, could authors comment on the implications of the results of their study on the everyday clinical practice?

Answer: As suggest by the reviewer, in the discussion section of the new manuscript, we have discussed this point. Unfortunately our study did not include non-hospitalized patients, considering only patients hospitalized; thus, the evaluation of liver function and damage tests at admission could be an useful marker to highlight patients at risk of COVID-19 progression in daily clinical practice.

We thank the Reviewers and the Editor for helping us to improve our paper.

We hope that the paper is now worthy of publication in Pathogens

Best regards,

Prof Nicola Coppola

Reviewer 2 Report

In this manuscript, the authors aim to evaluate the potential prognostic value of altered transaminases and bilirubin test results at the time of admission on the prognosis of patients diagnosed with COVID-19 in Italy. Using data from multiple centers, they show that patients with higher levels bilirubin or AST had a higher risk of poor COVID-19 related outcomes.

Overall, this study is interesting as it shows the potential prognostic value of liver tests in patients with COVID-19. However, there are some issues that remain to be addressed by the authors:

1-Since the authors are focused on the abnormality of liver tests as an indicator of poor outcomes, it would be more relevant to further analyze ALT and/or AST levels in terms of intervals of altered values (e.g. AST and/or ALT ranging between 2 and 4 x normal values ; 5x or more ; and 10x or more), especially considering that higher transaminases values usually indicate more severe liver disease in the acute setting.

2-Regarding the patients included in the current study: was COVID-19 required to be the principal condition of admission for inclusion in the current work, or were patients who presented with positive RT-PCR test while they were being admitted for other reasons and asymptomatic for COVID-19.

3-While the current manuscript is interesting, it would be more valuable to predict the mortality risk based on altered liver tests using dedicated predictive models and construct potentially a nomogram accordingly.

4-The results of the ordinal logistic regression show a significant but extremely minor association for higher serum AST with severe clinical outcomes, with an OR of 1.002. The clinical interpretation of such findings remains very unclear and in need of more validation using additional cohorts of patients.

5-Did the authors try to control for patients (who may not have a chronic liver disease) who have altered liver tests in the baseline (non-COVID-19) setting (e.g. patients who are known to have altered liver tests >1 week before COVID-19 diagnosis)?

6-The authors assess AST, ALT and bilirubin values at admission and evaluate their potential association with COVID-19 outcomes. However, they should control for patients who have pre-existing liver conditions that could explain abnormal liver test results independently of COVID-19, by including chronic liver disease as a potential confounding factor in the multivariate analysis.

7-Were normal ranges for AST and bilirubin the same across all COVID-19 units included, or were abnormal values defined differently? What were specifically the normal ranges for AST and bilirubin considered? These details need to be added to the manuscript.

Author Response

Dear Editor,

We re-submit our paper “Prognostic value of transaminases and bilirubin levels at admission to hospital on disease progression and mortality in patients with covid-19 - an observational retrospective study (tracking number: pathogens-1704753), modified according to the suggestions of the Editor and of the Reviewers.

POINT-BY-POINT ANSWER TO THE COMMENTS OF THE REVIEWER 2

Point 1: Since the authors are focused on the abnormality of liver tests as an indicator of poor outcomes, it would be more relevant to further analyze ALT and/or AST levels in terms of intervals of altered values (e.g. AST and/or ALT ranging between 2 and 4 x normal values ; 5x or more ; and 10x or more), especially considering that higher transaminases values usually indicate more severe liver disease in the acute setting.

Answer: Following the suggestion of the reviewer, we have included the data required in the Result section and as Figure 2 of the new manuscript.

Point 2: Regarding the patients included in the current study: was COVID-19 required to be the principal condition of admission for inclusion in the current work, or were patients who presented with positive RT-PCR test while they were being admitted for other reasons and asymptomatic for COVID-19.

Answer: Following the suggestion of the reviewer, we have specified that all the patients enrolled in the present paper were symptomatic for COVID-19

Point 3: While the current manuscript is interesting, it would be more valuable to predict the mortality risk based on altered liver tests using dedicated predictive models and construct potentially a nomogram accordingly.

Answer:  Thanks for the useful and proper suggestion, but considering the needs of a validation group and the further analysis to be carried out, we will perform this analysis in a subsequent study.

Point 4: The results of the ordinal logistic regression show a significant but extremely minor association for higher serum AST with severe clinical outcomes, with an OR of 1.002. The clinical interpretation of such findings remains very unclear and in need of more validation using additional cohorts of patients.

Answer: Thanks for the suggestions. We have modified the manuscript accordingly (section DISCUSSION)

Point 5: Did the authors try to control for patients (who may not have a chronic liver disease) who have altered liver tests in the baseline (non-COVID-19) setting (e.g. patients who are known to have altered liver tests >1 week before COVID-19 diagnosis)?

Answer: Since the patients with a pre-existing chronic liver disease were only 63, no analysis has been possible on the impact of the presence of chronic liver disease on the outcome of COVID-19.

Point 6: The authors assess AST, ALT and bilirubin values at admission and evaluate their potential association with COVID-19 outcomes. However, they should control for patients who have pre-existing liver conditions that could explain abnormal liver test results independently of COVID-19, by including chronic liver disease as a potential confounding factor in the multivariate analysis.

Answer: We have modified the new manuscript according to the suggestion of the reviewer. Specifically, we have specified that we have excluded the patients with chronic liver disease from multivariate analysis considering the non-significant result of Chi square analysis (p= 0.053) and considering the total number of patients with chronic liver disease (63 patients), very low considering the total number of patients included (1,641).

Point 7: Were normal ranges for AST and bilirubin the same across all COVID-19 units included, or were abnormal values defined differently? What were specifically the normal ranges for AST and bilirubin considered? These details need to be added to the manuscript.

Answer: In the new manuscript, we have added these data.

We thank the Reviewers and the Editor for helping us to improve our paper.

We hope that the paper is now worthy of publication in Pathogens

Best regards,

Prof Nicola Coppola

Reviewer 3 Report

Thank you very much for the opportunity to read the manuscript entitled “PROGNOSTIC VALUE OF TRANSAMINASES AND BILIRUBIN LEVELS AT ADMISSION TO HOSPITAL ON DISEASE PROGRESSION AND MORTALITY IN PATIENTS WITH COVID-19 - AN OBSERVATIONAL RETROSPECTIVE STUDY”.

This paper provides well described and structured data from a large number of patients form a multicentre group. Data are interesting but do not provide very novel information (authors include the references [8-23]). Despite the interest of confirming previously reported data, it does not provide very useful data, as the use of liver function test guiding management is tricky cause it may be influenced by many other issues (and actually the group of mild clinical outcome has higher AST and ALT than the moderate one), and the conclusion is vague without any specific indication. Nevertheless, whether performing statistical analysis, authors could indicate cut-off values for blood tests indicated to differentiate risk groups, the manuscript could have enough interest to be published.

Major comments:

  • Were all patients admitted due to respiratory symptoms? Any other reasons for admission? Were all cases of death attributed to SARS-CoV-2?
  • Were all RT-PCR the same across the different sites? Any differences (brand, threshold for positivity…)? Please, provide information or at least consider including a statement similar to “… on naso-oropharyngeal swab, according to indications by manufacturer, from… “
  • Follow-up of patients: patients may have a negative result after initial positive, and die after this switch due to pulmonary complications or others. Do authors have data regarding death during admission after negativity?
  • No differences in patients from different groups related to liver chronic disease, please provide discussion about it
  • Consider potential relation with different variants across the period of study (21 months…), this may interfere (limitations)
  • Discussion about potential different treatments in the patients along the time of the study, how it may impact in the results/outcomes observed
  • Discuss about higher ALT and AST in group “mild clinical outcome” in comparison with “moderate”
  • If authors want to provide a tool, please suggest a threshold of transaminases and bilirubin to guide management

Minor comments:

  • Line 157: “the four 156 groups
  • Increase size of font from grey box of Figure 1
  • Review IQR of “Days from admission to discharge” from Table 1
  • Provide units for blood tests (leukocytes and biochemistry values) in Table 1
  • Consider full stop in line 234: “ …cells. However …”

Author Response

Dear Editor,

We re-submit our paper “Prognostic value of transaminases and bilirubin levels at admission to hospital on disease progression and mortality in patients with covid-19 - an observational retrospective study (tracking number: pathogens-1704753), modified according to the suggestions of the Editor and of the Reviewers.

POINT-POINT ANSWER TO THE COMMENTS OF THE REVIEWER 3

Point 1: Were all patients admitted due to respiratory symptoms? Any other reasons for admission? Were all cases of death attributed to SARS-CoV-2?

Answer: As required by the reviewer, in the new manuscript we have clarified that all patients included in the cohort were symptomatic for COVID-19 and all deaths were to be considered related to direct or indirect complications of COVID-19

Point 2: Were all RT-PCR the same across the different sites? Any differences (brand, threshold for positivity…)? Please, provide information or at least consider including a statement similar to “… on naso-oropharyngeal swab, according to indications by manufacturer, from… “

Answer: Thanks for the question. This point has been clarified in the new manuscript.

Point 3: Follow-up of patients: patients may have a negative result after initial positive, and die after this switch due to pulmonary complications or others. Do authors have data regarding death during admission after negativity?

Answer: As required by the reviewer, this point has been clarified in the new manuscript.

Point 4: No differences in patients from different groups related to liver chronic disease, please provide discussion about it

Answer: Following the suggestion of the reviewer, we have analysed this point in the new manuscript. We thanks the reviewer for this suggestion.

Point 5: Consider potential relation with different variants across the period of study (21 months…), this may interfere (limitations)

Answer: In the present study, it was not evaluated the impact of viral variants. This limitation has been underlined in the new manuscript

Point 6: Discussion about potential different treatments in the patients along the time of the study, how it may impact in the results/outcomes observed

Answer: Following the suggestion of the reviewer, we have analysed this point in the new manuscript

Point 7: Discuss about higher ALT and AST in group “mild clinical outcome” in comparison with “moderate”

Answer: Thanks for your comment. Also according to the suggestions of the reviewer 4, we have changed the way to represent the data in Table 2 and 3. In particular, in table 2 the value of AST and ALT has been represented as Median (Q1-Q3), and not as mean. Looking the new representation, the difference showed in AST value between mild and moderate was no long identified and that in ALT value not significant.

Point 8: If authors want to provide a tool, please suggest a threshold of transaminases and bilirubin to guide management

Answer: Following the suggestion of the reviewer, we have performed a ROC curve for total bilirubin and AST. We considered two groups, Group 1 patients with mild or moderate disease and Group 2 patients with severe disease or dead. We founded that the AUC of total bilirubin at admission was 0.526 (95%CI 0.492-0.560) (p=0.129), and a cut off for severe outcome of 0.82 mg/dl (sensitivity: 30%, specificity 76%); for AST, the AUC was 0.570 (95%CI: 0.537-0.603) (p<0.001) and the cut off for severe outcome of 30.5 U/L (sensitivity: 60%, specificity: 53%). Considering the analysis performed above that don’t clearly support a threshold for AST and bilirubin and considering the analysis performed in the paper we support the hypothesis that the presence at admission of altered value of bilirubin and AST at admission could predict a worse prognosis.

Point 9: Line 157: “the four 156 groups”

Increase size of font from grey box of Figure 1

Review IQR of “Days from admission to discharge” from Table 1

Provide units for blood tests (leukocytes and biochemistry values) in Table 1

Consider full stop in line 234: “ …cells. However …”

Answer: We have modified the text according to the suggestions of the reviewer.

We thank the Reviewers and the Editor for helping us to improve our paper.

We hope that the paper is now worthy of publication in Pathogens

Best regards,

Prof Nicola Coppola

Reviewer 4 Report

In the introduction part of the article, it would be appropriate to write a few sentences about what the COVID-19 infection is and its epidemiology. For this purpose, you can benefit from the following studies:

BaÅŸkıran A, Akbulut S, Åžahin TT, Tunçer A, Kaplan K, Bayındır Y, Yılmaz S. Coronavirus Precautions: Experience of High Volume Liver Transplant Institute. Turk J Gastroenterol. 2022 Feb;33(2):145-152. doi: 10.5152/tjg.2022.21748.

Sahin TT, Akbulut S, Yilmaz S. COVID-19 pandemic: Its impact on liver disease and liver transplantation. World J Gastroenterol. 2020 Jun 14;26(22):2987-2999. doi: 10.3748/wjg.v26.i22.2987.

Altunisik Toplu S, Bayindir Y, Yilmaz S, Yalçınsoy M, Otlu B, Kose A, Sahin TT, Akbulut S, Isik B, BaÅŸkiran A, Koc C. Short-term experiences of a liver transplant centre before and after the COVID-19 pandemic.
Int J Clin Pract. 2021 Oct;75(10):e14668. doi: 10.1111/ijcp.14668.

All three tables should be revised as the way they were prepared. I recommend making a table in accordance with the APA style.
Table-2 and Table-3 should definitely be revised. It is very important for the reader's understanding. If I, as an epidemiologist and biostatistician, have difficulty understanding these charts, it is unlikely that other clinicians will.
A good article should be written clearly and simply, and use clear wording for the target audience to understand.
These tables should be checked by a biostatistician academic.

The analyzes of this work definitely need to be reinterpreted, and then I would like to reevaluate it.

Author Response

Dear Editor,

We re-submit our paper “Prognostic value of transaminases and bilirubin levels at admission to hospital on disease progression and mortality in patients with covid-19 - an observational retrospective study (tracking number: pathogens-1704753), modified according to the suggestions of the Editor and of the Reviewers.

POINT-POINT ANSWER TO THE COMMENTS OF THE REVIEWER 4

Point 1: All three tables should be revised as the way they were prepared. I recommend making a table in accordance with the APA style. Table-2 and Table-3 should definitely be revised. It is very important for the reader's understanding. If I, as an epidemiologist and biostatistician, have difficulty understanding these charts, it is unlikely that other clinicians will. A good article should be written clearly and simply, and use clear wording for the target audience to understand. These tables should be checked by a biostatistician academic.

Answer: As required by the reviewer, we have improved the text, added footnotes, to improve the reader's understanding, particularly in tables 2 and 3. As recommended, we have reviewed the tables by the help of a biostatistician, Dr. Vittorio Simeon.

Point 2: BaÅŸkıran A, Akbulut S, Åžahin TT, Tunçer A, Kaplan K, Bayındır Y, Yılmaz S. Coronavirus Precautions: Experience of High Volume Liver Transplant Institute. Turk J Gastroenterol. 2022 Feb;33(2):145-152. doi: 10.5152/tjg.2022.21748.

Sahin TT, Akbulut S, Yilmaz S. COVID-19 pandemic: Its impact on liver disease and liver transplantation. World J Gastroenterol. 2020 Jun 14;26(22):2987-2999. doi: 10.3748/wjg.v26.i22.2987.

Altunisik Toplu S, Bayindir Y, Yilmaz S, Yalçınsoy M, Otlu B, Kose A, Sahin TT, Akbulut S, Isik B, BaÅŸkiran A, Koc C. Short-term experiences of a liver transplant centre before and after the COVID-19 pandemic. Int J Clin Pract. 2021 Oct;75(10):e14668. doi: 10.1111/ijcp.14668.

Answer: Following the suggestion of the reviewer, we have update the references.

We thank the Reviewers and the Editor for helping us to improve our paper.

We hope that the paper is now worthy of publication in Pathogens

Best regards,

Prof Nicola Coppola

Round 2

Reviewer 3 Report

Some suggestions to consider before publication:

  • Instead of using “normal value”, I suggest: below “upper limit of normal” (ULN), and after “x1ULN, x2ULN…”
  • Line 283: “… it was not possible to evaluate…”
  • Again, increase font size of grey box from Figure 1
  • Line 180: “ … (Table 2). Moreover…”

Thank you.

Reviewer 4 Report

Dear Authors

Thank you for thispresentation. Revisions are enough for me